# Enhanced Angiogenesis in HUVECs Preconditioned with Media from Adipocytes Differentiated from Lipedema Adipose Stem Cells In Vitro

**DOI:** 10.3390/ijms241713572

**Published:** 2023-09-01

**Authors:** Sara Al-Ghadban, Samantha G. Walczak, Spencer U. Isern, Elizabeth C. Martin, Karen L. Herbst, Bruce A. Bunnell

**Affiliations:** 1Department of Microbiology, Immunology and Genetics, University of North Texas Health Science Center, Fort Worth, TX 76107, USA; samanthawalczak@my.unthsc.edu (S.G.W.); spencer.isern@unthsc.edu (S.U.I.); 2Department of Medicine, Section of Hematology and Oncology, Tulane University, New Orleans, LA 70118, USA; emartin1@tulane.edu; 3Total Lipedema Care, Tucson, AZ 85715, USA; kaherbst@gmail.com

**Keywords:** lipedema, endothelial cells, adipocytes, angiogenesis, notch signaling

## Abstract

Lipedema is a connective tissue disorder characterized by increased dilated blood vessels (angiogenesis), inflammation, and fibrosis of the subcutaneous adipose tissue. This project aims to gain insights into the angiogenic processes in lipedema using human umbilical vein endothelial cells (HUVECs) as an in vitro model. HUVECs were cultured in conditioned media (CM) collected from healthy (non-lipedema, AQH) and lipedema adipocytes (AQL). The impacts on the expression levels of multiple endothelial and angiogenic markers [CD31, von Willebrand Factor (vWF), angiopoietin 2 (ANG2), hepatocyte growth factor (HGF), vascular endothelial growth factor (VEGF), matrix metalloproteinase (MMPs), NOTCH and its ligands] in HUVECs were investigated. The data demonstrate an increased expression of CD31 and ANG2 at both the gene and protein levels in HUVECs treated with AQL CM in 2D monolayer and 3D cultures compared to untreated cells. Furthermore, the expression of the vWF, NOTCH 4, and DELTA-4 genes decreased. In contrast, increased VEGF, MMP9, and HGF gene expression was detected in HUVECs treated with AQL CM cultured in a 2D monolayer. In addition, the results of a tube formation assay indicate that the number of formed tubes increased in lipedema-treated HUVECs cultured in a 2D monolayer. Together, the data indicate that lipedema adipocyte-CM promotes angiogenesis through paracrine-driven mechanisms.

## 1. Introduction

Increased blood vessel formation, intercellular fibrosis, hypertrophic adipocytes, and immune cell infiltration are significant hallmarks of lipedema, a painful subcutaneous adipose tissue disorder [1,2,3,4]. Previous data demonstrated that approximately 30% of women with lipedema had increased angiogenesis in the thigh adipose tissue, resulting in dilated and leaky capillaries, apoptotic/necrotic adipocytes, collagen deposition, and interstitial fluid accumulation [1]. Herbst and colleagues showed that women with lipedema have more leaky blood vessels with abnormal phenotype and increased dermal space in thigh skin compared to non-lipedema controls, indicating dysfunctional micro-blood vessels [5]. In addition, Crescenzi demonstrated an accumulation of tissue sodium in women with lipedema associated with increased inflammation and endothelial barrier disruption [6]. These data suggest angiogenesis in lipedema tissue is associated with vascular remodeling and alterations in the extracellular matrix (ECM).

Angiogenesis in adipose tissue is a tightly regulated process that involves (1) the proliferation and migration of endothelial cells (ECs), (2) cellular interactions between ECs, pericytes, immune cells, and adipocytes, (3) cell–matrix interactions, and (4) secretion of pro- and anti-angiogenic and growth factors such as vascular endothelial growth factor (VEGF), fibroblast growth factors (FGF), matrix metalloproteinases (MMP-2 and MMP-9), interferon (IFN-α and -β), and angiopoietin [7,8,9,10]. However, under pathological conditions, such as obesity and cancer, the balance between pro- and anti-angiogenic factors is altered, leading to an increase in pathological angiogenesis, inflammation induction, and ECM disruption, which results in endothelium dysfunction [11,12,13,14,15,16].

Studies have shown that endothelial dysfunction is associated with EC activation, as indicated by the expression of cell surface markers, including vascular cell adhesion molecule-1 (VCAM-1), intercellular adhesion molecule-1 (ICAM-1), and platelet endothelial cell adhesion molecule (PECAM-1; CD31), secretion of angiopoietin-2 (Ang-2) and von Willebrand factor (vWF) [13,14,17]. Furthermore, EC activation is induced by pro-inflammatory cytokines, such as interleukins (IL-1β, IL-6) and tumor necrosis factor (TNF-α), which are produced primarily by recruited monocytes and activated macrophages [18,19,20]. In addition, preadipocytes and adipocytes synthesize and release various pro-inflammatory and anti-inflammatory factors, including adipokines, cytokines, chemokines, and growth factors. These also contribute to the EC dysfunction observed in obese adipose tissue [21,22,23,24,25]. Thus, the crosstalk between the ECs and adipocytes, mediated through direct cell–cell communication and via secreted molecules, plays a crucial role in instigating pathological angiogenesis in adipose tissue disorders, such as lipedema.

Several studies have demonstrated an increase in the levels of angiogenic and inflammatory cytokines and chemokines in the serum of lipedema patients, including VEGFA and VEGFC, transforming growth factor (TGFα and β1), interleukin 8 (IL-8), chemokines C-X-C motif chemokine (CXCL3 and 11), and monocyte-chemotactic protein (MCP-1 and -3) [26,27,28,29]. In addition, the observed increase in the number of activated macrophages contained in both the stroma vascular fraction (SVF) [30] and adipose tissue [1,3,26] isolated from lipedema patients contributes to the pathological angiogenesis in lipedema by inducing endothelial dysfunction and altering vascular permeability through the secretion of pro-inflammatory cytokines.

Therefore, to gain insights into lipedema-associated angiogenesis, adipose-derived stem cells (ASCs) isolated from the SVF of lipedema and non-lipedema (healthy) donors were differentiated into adipocytes in vitro, conditioned media (CM) were collected, and their effects tested on the angiogenesis of human umbilical vein ECs (HUVECs) in vitro. In the current study, the expression of multiple angiogenic markers expressed by HUVECs was analyzed at both the transcriptional and translational levels in 2D monolayer and 3D cultures. The data demonstrated the increased expression of the primary angiogenic markers, CD31, ANG2, VEGF, and MMP9, and a decrease in vWF, NOTCH (1–4), and DELTA-4 in HUVECs treated with lipedema adipocyte-CM as compared to the control untreated HUVECs in the 2D monolayer culture. Likewise, an increase in the expression of CD31 and ANG2 was detected in HUVEC 3D scaffolds treated with lipedema adipocyte-CM, with no change in VEGF expression.

## 2. Results

### 2.1. Preconditioning of HUVECs with Adipocyte Media Increases the Expression of CD31 and ANG2 in 2D Monolayer Culture

To determine the effect of the conditioned media collected from healthy and lipedema pooled cells on the angiogenic potential of HUVECs, cells were treated for 24 h and then analyzed by qRT-PCR and Western blot for the expression of key angiogenic genes and proteins. The transcriptional analysis demonstrated a significant upregulation of *CD31* (2- and 4-fold) and *ANG2* (~5-fold) in HUVECs treated with AQH and AQL CM, respectively, compared to ASCs CM-treated and untreated control cells (UnTx), suggesting the induction of angiogenesis. It is worth noting that there is also a significant 2-fold increase in the expression of *CD31* in HUVECs treated with AQL CM compared to AQH-treated cells, suggesting the differential effect of AQL CM on the induction of angiogenesis in HUVECs (Figure 1A,B). In addition, Western blot analysis revealed a significant 2-fold increase in the protein levels of ANG2 in HUVECs treated with ASCL and AQH CM and a 5-fold increase in AQL CM compared to control cells (Figure 1C,E, Appendix A). CD31 protein levels were also increased when treated with AQL CM; nonetheless, they did not reach significance (Figure 1C,D, Appendix A). However, flow cytometry detected a significant 2-fold increase in CD31 cell surface marker expression in AQ-treated cells compared to UnTx cells (Figure 1F) with no change in the number of cells expressing CD31 (Appendix A). A 2-fold increase in CD31 cell surface expression was also detected in HUVECs treated with ASC CM.

### 2.2. Preconditioning of HUVECs with Adipocyte Media Increases the Expression of Angiogenic, Inflammatory, and ECM Markers in 2D Monolayer Culture

The expression of 84 genes involved in modulating the biological processes of angiogenesis was analyzed in HUVECs treated with different CM collected from pooled healthy and lipedema cells using the human angiogenesis RT^2^ Profiler PCR array. The data showed a significant increase in angiogenic markers (ANG, ANGPT1, ANGPT2, ANPEP, and HGF) in HUVECs treated with AQH and AQL CM compared to ASC CM-treated and untreated cells (Figure 2A), confirming the data presented in Figure 1. In addition, the data also demonstrated an increase in the expression of numerous inflammatory markers [*VEGFA, VEGFB, VEGFC, VEGFD, HIF1A, (VEGFR1),* and *KDR (VEGFR3*)] and ECM markers (*ANGPTL4, LEP, MMP2, MMP9, MMP14*) (Figure 2A) in AQL HUVEC-treated cells compared to untreated cells. HUVECs were treated with CM collected from undifferentiated ASCs and adipocytes differentiated from healthy and lipedema ASC lines to validate the results obtained from the PCR array. The data demonstrated a significant increase in the expression of *MMP9* (2.5-fold), *leptin* (5-fold), and *HGF* (4-fold) in HUVECs treated with AQL CM compared to untreated cells (Figure 2C–E). Additionally, an increase in *MMP9* (~2-fold, ASCL CM), *leptin* (2.5-fold ASCH CM; ~3-fold ASCL CM), and *HGF* gene expression (~5-fold ASCH CM) was detected in HUVECs treated with ASC CM; however, they did not reach significance. Interestingly, HUVECs treated with ASCL CM showed a 2-fold increase in the expression of the *HGF* gene compared to untreated cells, suggesting that CM media from lipedema ASCs might instigate angiogenesis (Figure 2A,E). No changes were detected in the expression of *MMP2* in treated cells (Figure 2B). However, a significant decrease in the gene expression of adherens junction, *VE-cadherin* in ASL and AQL-treated cells (~0.5-fold decrease, Figure 2F), and gap junction, *Cx43*, in HUVECs treated with ASCH and AQ CM was detected compared to untreated cells (~0.6-fold decrease, Figure 3A).

Furthermore, the decrease in Cx43 gene expression (Figure 3A) was not associated with reduced dye transfer in HUVECs (Figure 3B,C). Treated HUVECs with CM demonstrated a ~2-fold increase in intercellular communication compared to untreated cells. Interestingly, the percentage of calcein-labeled recipient-treated HUVECs was significantly decreased compared to untreated recipient cells (Appendix A), which might be associated with the decrease in VE-cadherin gene expression shown in Figure 2F, thus suggesting a reduction in cellular adhesion in treated HUVECs.

### 2.3. Preconditioning of HUVECs with Adipocyte Media Decreases the Expression of vWF in 2D Monolayer Culture

Next, we investigated the effect of CM on the expression of the primary regulator of the angiogenic pathway, von Willebrand Factor (vWF). A significant decrease in *vWF* gene expression (<0.5-fold) was demonstrated in HUVECs treated with ASCL, AQH, and AQL CM collected from single cells compared to ASCH CM-treated and untreated control cells (Figure 4A). It is worth noting that a significant 2-fold decrease in *vWF* gene expression was detected in HUVECs treated with AQ CM compared to ASCH CM-treated cells.

Furthermore, at the protein level, vWF expression was decreased in HUVECs treated with ASCH CM (0.5-fold) and AQL CM (0.7-fold) compared to untreated cells (Figure 4B and Appendix A), confirming the paracrine effect of AQL CM on inducing angiogenesis.

### 2.4. Preconditioning of HUVECs with Adipocyte Media Decreases the Expression of NOTCH and Its Ligands in 2D Monolayer Culture

To further examine the effect of CM on the induction of angiogenesis in HUVECs, we investigated a panel of genetic components of the notch signaling pathway using the RT^2^ Profiler™ PCR Array Human Notch Signaling Pathway Plus. The data showed a decrease in the expression of *NOTCH* [(*NOTCH1*, *NOTCH2*, *NOTCH3*, *NOTCH4*)] in HUVECs treated with AQH and AQL CM compared to ASC CM-treated and untreated cells (Figure 5A). The PCR array also demonstrated a decrease in *Delta-4* and *Jag-1* in HUVECs treated with ASCH and ASCL CM compared to AQ CM-treated and untreated cells (Figure 5A). We validated the results obtained from the PCR array by performing qPCR on HUVECs treated with CM collected from stem cells and differentiated adipocytes from healthy and lipedema single cells. The data demonstrated a significant decrease in *NOTCH2* gene expression in AQ CM-treated cells (~0.5-fold) compared to ASC CM-treated and untreated cells (Figure 5B) and a remarkable decrease in *NOTCH4* gene expression in all HUVEC-treated cells compared to untreated cells (<0.5-fold; Figure 5C). A significant decrease in *Delta-4* and *Jag-1* in HUVECs treated with ASCH and ASCL CM was also detected compared to AQ CM-treated and untreated cells (<0.3-fold; Figure 5D; 0.6-fold; Figure 5F). The PCR array also demonstrated a significant increase in the inflammatory markers (*NF-κB1* and *2, TNFSF10*, and *IFNG*) in AQL CM-treated cells compared to untreated cells aligned with the data presented in Section 2.2 (Figure 2A).

### 2.5. Increase Tube Formation in Preconditioned HUVECs in 2D Monolayer Culture

To investigate the angiogenic effects of CM on HUVECs, a tube formation assay was performed. HUVECs cells were cultured in either EGM-2 (control cells), ASC-CM, or AQ-CM for 18 h (Figure 6A). Quantification of the tube formation showed a significant 4.8- and 6.5-fold increase in HUVECs treated with ASCL and AQL CM, respectively, compared to untreated control cells (Figure 6B). It is worth noting that HUVECs treated with CM from ASL and AQL showed a significant 2-fold increase compared to HUVECs treated with CM from ASH and AQH, respectively, confirming that lipedema CM induce endothelial tube formation.

### 2.6. Preconditioning of HUVECs with Adipocyte Media Increases the Expression of Angiogenic Markers in 3D Culture

Studies have demonstrated that cells grown in 3D cultures more accurately duplicate the in vivo microenvironment; thus, the response to treatment (CM or drugs) might be different compared to cells grown in 2D monolayer cultures due to the spatial arrangement of the cells and cell–cell and the cell–ECM interactions [31,32,33]. Furthermore, studies have shown that HUVECs cultured in 3D demonstrated solid and firm vascular-like networks, specifically in a co-culture with either mesenchymal cells or ASCs [34,35,36,37]. Thus, the effect of CM on the angiogenic potential of HUVECs in 3D cultures was assessed after examining the effect of CM on HUVECs in 2D monolayer cultures. HUVECs cultured in 3D scaffolds were treated with the identical CM from pooled cells, as mentioned in the methods, for 48 h and then analyzed by qRT-PCR and Western blot for the expression of CD31 and ANG2. The transcriptional analysis demonstrated a significant 3-fold upregulation of *CD31* expression in HUVECs treated with AQL CM compared to ASC CM-treated and untreated control cells (Figure 7A). AQH CM-treated cells also showed a ~2-fold increase as compared to untreated cells; however, this increase is not statistically significant. Furthermore, *ANG2* gene expression showed a significant 2-fold increase in AQ CM-treated cells compared to untreated control cells (Figure 7B). It is worth mentioning that HUVECs treated with ASC CM showed an increase in *ANG2* expression, indicating that ASCs CM have the potential to instigate angiogenesis in HUVEC 3D cultures. The difference demonstrated in the *ANG2* gene expression in HUVECs treated with ASCs CM between 2D monolayer and 3D cultures is mainly due to the difference in the culture conditions, including the cellular matrix, the microenvironment, and the effect of the treatment.

Western blot analysis revealed increased CD31 protein levels in treated cells; however, only cells treated with ASCL CM reached significance (1.5-fold, *p* < 0.001; Figure 7C,D, Appendix A). ANG2 protein levels were significantly increased in HUVECs treated with AQL CM compared to untreated control cells (3-fold, *p* < 0.01; Figure 7C,E, Appendix A).

## 3. Discussion

Endothelial cell activation, matrix degradation, increased vascular permeability, and recruitment of inflammatory cells induce abnormal angiogenesis in the context of pathological diseases, such as obesity and cancer [38,39,40]. In addition to that, the cellular interactions between the SVF and adipocytes promote tissue angiogenesis through direct and indirect paracrine mechanisms. Paracrine signaling molecules, secreted by ASCs and adipocytes, contain a plethora of pro- and anti-inflammatory cytokines, chemokines, and angiogenic and growth factors that have been shown to play a crucial role in several processes, including wound healing, cellular differentiation, tissue repair and regeneration, and angiogenesis [41,42,43,44,45,46,47,48].

The present study assessed the effect of conditioned media (CM) collected from lipedema and healthy adipocytes on the angiogenic properties of ECs (HUVECs). The data show the increased expression of the endothelial marker, CD31, and the angiogenic marker, ANG2, in HUVECs treated with CM from healthy and lipedema ASCs and differentiated adipocytes in 2D monolayer and 3D cultures, suggestive of angiogenic induction in vitro. It is worth noting that the effect of CM on HUVECs was not transient. The increase in the gene expression of CD31 and ANG2 in treated HUVECs was maintained in a 2D monolayer culture for 72 h after the CM were removed (Appendix A).

Studies have also shown that angiopoietins (ANG1 and ANG2) and growth factors (VEGF and FGF) act synergistically to induce angiogenesis in several diseases [8,49,50,51,52]. Our data show an upregulation in the gene expression of VEGF (A–D), VEGFR1 (FLT1), VEGFR2 (KDR), FGF (1–2), and HGF in HUVECs treated with CM from lipedema adipocyte as compared to untreated and HUVECs treated with CM from healthy adipocytes and ASCs, suggesting that the CM of lipedema adipocytes promote angiogenesis in HUVECs.

Furthermore, HUVECs treated with ASCL CM showed the increased expression of the HGF gene compared to ASCH CM-treated and untreated cells, suggesting that CM media from lipedema ASCs might also instigate angiogenesis. These data are confirmed by the significant increase in the gene expression of the major ECM markers, MMP9, angiopoietin-like 4 (ANGPTL4), and leptin in HUVECs treated with AQL CM compared to untreated and HUVECs treated with CM from healthy cells. ANGPTL4 expression is induced by hypoxia which has been documented in lipedema adipose tissue due to dysregulated microvasculature, inflammation, and increases in interstitial fluid [1,4,5,53], and in cancer such as in uveal melanoma [54]. Furthermore, analysis of the RNA-seq data through Gene Ontology (GO) pathway enrichment analysis demonstrated an altered gene expression associated with angiogenesis and vascular development in lipedema ASCs compared to healthy cells (Appendix A). Focusing on the angiogenesis-controlled pathway, a decrease in the genes associated with an increase in angiogenesis in lipedema ASCs was demonstrated. Specifically, FLT1 was observed to be downregulated (Appendix A). Studies conducted by Chappell et al. demonstrated that FLT-1 regulates blood vessel formation via the VEGF-NOTCH signaling pathway [55,56]; also, Bosco et al. showed that the loss of FLT-1 disrupts these signaling dynamics and results in vessel dysmorphogenesis [57]. Thus, our data suggest that angiogenesis in lipedema is probably induced in progenitor/stem cells before their differentiation into ECs.

Leptin, a major adipokine marker secreted by adipocytes, regulates angiogenesis by inducing EC proliferation, tube formation, and the expression of MMPs [58,59]. Our data show an increase in the number of tubes formed by HUVECs treated with both healthy and lipedema ASCs and adipocytes compared to untreated HUVECs in a 2D monolayer culture; however, CM collected from lipedema ASCs and differentiated adipocytes demonstrated a remarkably higher angiogenic effect compared to CM collected from healthy cells. We also show a significant downregulation in vWF expression, an angiogenic marker associated with endothelium dysfunction in treated HUVECs. A decrease in the adherens junction VE-cadherin and Cx43 expression in HUVECs treated with CM from lipedema ASCs and adipocytes was demonstrated, linking angiogenesis with cell–matrix degradation, leading to a decrease in cellular adhesion but not intercellular communication. Our data are consistent with a study conducted by Strohmeier et al. 2022 that showed a decrease in endothelial adherens (VE-cadherin) and tight junction (ZO-1) gene expression in healthy human primary ECs treated with CM derived from lipedema SVF, confirming the increase in the endothelial permeability and leakiness of ECs; thus, endothelial barrier dysfunction [60]. Furthermore, studies have also shown that vWF controls the levels of ANG2 secreted by activated ECs via VEGFR2, resulting in blood vessel disruption and angiogenesis [10,15], consistent with the phenotype detected in lipedema tissue.

Notch signaling plays a crucial role in angiogenesis. Notch proteins and their receptors (Delta1–4) and ligands (Jagged 1–2) are induced by leptin and regulated by VEGF signals [61]. Our data show a decrease in NOTCH 1–4 gene and Delta 1 and 4 in HUVECs treated with CM from lipedema adipocytes compared to untreated and treated HUVECs with CM from healthy adipocytes. Furthermore, this decrease in Notch gene expression is associated with increased VEGF-A-D, NF-kb, and IFN-γ gene expression in HUVECs treated with CM from lipedema adipocytes, confirming that the induction of pathological angiogenesis is associated with inflammation in lipedema.

Combining all of these data, it is clear that healthy and lipedema ASCs and adipocyte-CM promote angiogenesis through paracrine-driven mechanisms. However, lipedema adipocyte-CM demonstrated a significantly higher effect on the HUVECs’ angiogenic potential than healthy CM. In addition, 3D cultures of HUVECs were developed and the data indicated that the lipedema adipocyte-CM increased the expression of CD31 and ANG2 in treated HUVECs at both the transcriptional and translational levels.

The following steps will be (1) performing in-depth characterizing of the CM collected from healthy and lipedema-differentiated adipocytes to determine the central cytokines that are driving angiogenesis and (2) determining the effect of direct co-culture lipedema ASCs/adipocytes with HUVECs on the expression of angiogenic markers, new tube formation, and EC activation. Developing a 3D model replicating the adipose niche will provide deeper insights into the ECM-vascular remodeling in lipedema tissue. Furthermore, this model will be used to screen potential drugs targeting the angiogenic markers, such as VEGF, ANG2, and MMP9, which will help reduce adipocyte hypertrophy and the formation of dysfunctional microvasculature, thereby providing new treatment opportunities for lipedema.

This study has two main limitations: (1) the lipedema samples used in this study were limited to Stage 2 due to the limited availability of ASCs from other stages of the disease, and (2) the ECs used are HUVECs and not primary ECs isolated from lipedema patients. Therefore, our future studies will include all lipedema stages to fully delineate the events from the early initiation (Stage 1) and progression, respectively. Furthermore, ECs from lipedema and non-lipedema SVF will be isolated and grown, which has been difficult for us to accomplish due to the lack of primary tissue samples.

## 4. Materials and Methods

### 4.1. Cell Culture

Human umbilical vein endothelial cells (HUVECs), from pooled donors (C2519A, Lonza, Walkersville, MD, USA). HUVECs were cultured in EGM-2 (endothelial growth medium 2; CCM027, R&D, Minneapolis, MN, USA) and used until passage 8. Adipose-derived stem cells (ASCs) were maintained in Dulbecco’s Modified Eagles Medium (DMEM)/F12 (Hyclone, Logan, UT, USA) supplemented with 10% heat-inactivated fetal bovine serum (FBS, Hyclone, Logan, UT, USA) and 1% antibiotic/antimycotic (ThermoFisher Scientific, Waltham, MA, USA). ASC cell lines, isolated from lipoaspirates of subcutaneous adipose tissue from the thigh, were fully characterized individually in our previous paper before being pooled [62]. ASCs were used as single cell lines or pools of cells from 5 separate donors. ASCs were used in passages 4–6 for the experiments. Table 1 summarizes the biological characteristics of ASCs used in this study.

### 4.2. Adipogenic Differentiation

ASCs were seeded at 2 × 10^4^ cells/cm^2^ density and grown in DMEM/F12. At confluence, the medium was replaced with adipogenic differentiation-inducing medium (AdipoQual, Obatala Biosciences, New Orleans, LA, USA) for the differentiation of the cells or kept in DMEM/F12 medium for control undifferentiated cells. Media were changed every 3 days until day 21 (D21). Conditioned media (CM) from control undifferentiated cells and differentiated cells were collected at D21, centrifuged at 180× *g* for 5 min to remove the debris, aliquoted, and stored at −80 °C for further experiments. Healthy and lipedema cell lines were seeded for all experiments at the same cell density and passage number.

### 4.3. Treatment with CM

HUVECs were seeded at 1 × 10^4^ cells/cm^2^ density in EGM-2 media and were cultured to 80–90% confluence in six-well plates. The cells were then washed 2× with 1× phosphate-buffered saline (PBS), and then treated with CM collected from healthy and lipedema ASCs (ASCH CM and ASCL CM, respectively) and healthy and lipedema-differentiated adipocytes (AQH CM and AQL CM, respectively) or left untreated (UnTx) for 24 h. Cells were then collected for flow cytometry, RNA, and protein assays.

### 4.4. Flow Cytometry

HUVECs, controlled and treated, were collected at 24 h and stained with FITC anti-human CD31 Antibody (Cat# 303104, BioLegend, San Diego, CA, USA). The cells were then fixed with 1% paraformaldehyde (PFA) and a total of 10,000 events were captured and analyzed with BD Accuri C6 plus (BD, Franklin Lakes, NJ, USA).

### 4.5. Dye Transfer Assay

HUVECs were seeded in a 100 mm^2^ Petri dish and labeled with 0.25 μM calcein-AM (Invitrogen, Thermo Fisher Scientific, Waltham, MA, USA), washed in serum-free medium for 15 min, and then washed and incubated with complete medium for 15 min to allow intercellular esterase to convert non-fluorescent calcein-AM to a green fluorescent calcein. Labeled HUVECs were co-cultured with unlabeled HUVECs treated with different CM as detailed in 2.3 for 30 min at 37 °C and 5% CO_2_. Non-adherent cells were removed by washing and the adherent cells were detached by trypsinization and resuspended in a complete medium containing 2% formaldehyde to be analyzed via flow cytometry. Dye transfer was determined by measuring the mean fluorescence intensity (MFI) and data were calculated by dividing MFI over the percentage of adhered fluorescent cell population after each co-culture (Appendix A)

### 4.6. HUVECs 3D Scaffolds

HUVECs (p5) were resuspended in 15% GelMA, as described previously [63], fibrinogen, and thrombin (Sigma-Aldrich, St. Louis, MO, USA) solution at 5 × 10^6^ cells/mL at a ratio 1:1:4. The gelation was photoactivated using a dental curing light (395 nm wavelength; LED wholesalers, Hayward, CA, USA) for 2 min. Then, scaffolds were placed in static culture in 12-well plates and treated with ASCs and AQ CM, as mentioned in Section 2.3, for 48 h. After treatment, scaffolds were collected for RNA and protein assays.

### 4.7. Quantitative Polymerase Chain Reaction (qPCR)

Total RNA from HUVECs was extracted using an RNA extraction kit (Qiagen, Germantown, MD, USA). A total of 1µg of mRNA was used for cDNA synthesis with an Applied Bioscience purification kit (Thermo Fisher Scientific, Waltham, MA, USA). qRT-PCR was performed using the SYBR Green qPCR SuperMix (Bio-Rad, Hercules, CA, USA) according to the manufacturer’s instructions. Oligonucleotide primers were designed with the vendor’s software (IDT, Newark, NJ, USA; https://www.idtdna.com/SciTools, accessed on 12 December 2018). Table 2 lists the primer sequences used for qRT-PCR. PCR conditions were: 2 min at 95 °C and 40 cycles of 15 s at 95 °C and 30 s at 60 °C. The target and reference genes were amplified in separate wells. All reactions were performed in duplicate. The 2^−∆∆CT^ method was used to quantify gene expressions and data were normalized to GAPDH, which was used as an internal control.

### 4.8. RT2 Profiler PCR Arrays

To screen for angiogenic markers, we used RT^2^ Profiler™ PCR Array Human Angiogenesis (Cat#: PAHS-024Z, Qiagen, Germantown, MD, USA) and RT^2^ Profiler™ PCR Array Human Notch Signaling Pathway Plus (Cat#: PAHS-059Y). Each array contains 86 genes, 5 housekeeping genes, human genomic DNA contamination, reverse transcription, and positive PCR controls. Total RNA from HUVECs (control and treated) was extracted using an RNA extraction kit (Qiagen). A total of 0.5 µg of mRNA was used for cDNA synthesis using the RT^2^ First Strand Kit (Cat# 330401, Qiagen). cDNA was then mixed with 2 × RT2 SYBR Green qPCR Master Mix and RNase- and DNase-free water and loaded onto the plates following the manufacturer’s instructions. RT-PCR was performed with a CFX96 thermocycler (Bio-Rad, Hercules, CA, USA) All the samples passed the quality control checks using the GeneGlobe Program provided by Qiagen (https://geneglobe.qiagen.com/us/analyze, accessed on 30 January 2023). RT2 PCR array data were normalized against the housekeeping genes (GAPDH, ACTB, B2M, RPLP0, and HPRT1) by calculating the 2^−∆∆CT^ for each gene of interest in the plate. Heatmaps were generated and analyzed by using GeneGlobe Program. Finally, the candidate genes were chosen to be validated in an additional qRT-PCR experiment.

### 4.9. Western Blot Analyses

Capillary Western analyses were performed using the ProteinSimple Jess System. HUVECs were lysed with RIPA lysis buffer (Cat #: 89900; Thermo Fisher) supplemented with 1× protease inhibitor (Cat #: 1862209; Thermo Fisher). Protein samples were quantified by using the bicinchoninic acid assay (BCA, Cat #: 23225; Thermo Fisher), and a total of 0.4 mg of protein lysate was loaded onto the plate along with the following primary antibodies for CD31 (1:1000; Cat #:MA5-32126; Thermo Fisher), angiopoietin 2 (1:150; Cat #: ab155106; Abcam, Cambridge, UK), vWF (1:100; Cat #: ab154193, Abcam), vinculin (1:500; Cat #: ab155120), and GAPDH (1:300; Cat #: 2118; Cell Signaling Technologies). After loading the plate according to the manufacturer’s instructions, the separation electrophoresis and immunodetection steps occurred in the fully automated capillary system. Jess Western data were analyzed using Compass for Simple Western software (Version 6.2.0; ProteinSimple, Bio-Techne, Minneapolis, MN, USA). The area under curves from chemiluminescence chromatograms was used to determine the relative amount of proteins. Expression levels of all proteins were normalized to GAPDH for 12–230 kDa Separation Modules and vinculin for 66–440 kDa Separation Modules (ProteinSimple, Bio-Techne, Minneapolis, MN, USA). Pseudo-blots, generated by the compass software from the high dynamic range 4.0, were presented with each protein of interest.

### 4.10. Matrigel Tube Formation Assay

HUVECs were resuspended in either EGM-2 media (UnTx) or CM (ASCH, ASCL, AQH and AQL) and seeded at a density of 60 × 10^3^ cells/cm2 onto a pre-incubated Matrigel plate (Corning, NY, USA). The cells were incubated at 37 °C and 5% CO_2_ for 18 h. Tube-like structures were determined by phase-contrast microscopy (20× magnification, EVOS Cell Imaging Systems, Thermo Fisher). The number of tubes formed were manually quantified using ImageJ software (Https://Imagej.Nih.Gov/Ij/, accessed on 30 January 2023).

### 4.11. Next-Generation RNA Sequencing

RNAseq libraries were prepared using TruSeq Stranded Total RNA Library Prep kits (Illumina, Cat #: 20020598) following the manufacturer’s instructions. Briefly, one microgram of total RNA was used as starting material. The ribosomal RNA was depleted using the RiboZero Gold included in the kit. The resulting RNA samples were fragmented and converted to double-stranded DNA for sequencing. Individual libraries were uniquely indexed using TruSeq RNA UD Indexes (Illumina, Cat #: 20020591) and pooled in an equimolar ratio. The pooled libraries were sequenced on an Illumina NextSeq 550 sequencing system for 150-bp paired-end reads. The analysis was performed using the “RNA Express” app that uses the STAR aligner to map the sequence reads to the reference genome Genome Reference Consortium Human Build 37 (GRCh37, also known as hg19) and quantified using the DESeq2 software (Version 1.1.0) in the back end. The raw sequence data files were deposited in the Sequence Read Archive (SRA) with reference number PRJNA946154. Unbiased pathway analysis was performed through ENRCHR Ontologies, GO: Biological Processes data set [64,65,66]. Genes included in the analysis were all significantly changed genes with an adjusted *p*-value of *p* < 0.05.

### 4.12. Statistical Analysis

GraphPad PRISM 8 was used for all statistical analyses. Mann–Whitney test was used to determine the differences between the two groups of participants. One-way ANOVA followed by Tukey’s post hoc test were used to analyze the differences between the four groups. Asterisks (*) indicate statistical significance: * *p* < 0.05; ** *p* < 0.01; *** *p* < 0.001; **** *p* < 0.0001.

## 5. Conclusions

The data presented here indicate that angiogenesis associated with lipedema is partially regulated by paracrine signaling mediated by growth factors produced by progenitor/stem cells and adipocytes. Understanding the molecular mechanisms that induce and regulate angiogenesis in lipedema adipose tissue is vital to give insights into the pathology of the disease and for the development of novel approaches for treatment.

## Figures and Tables

**Figure 1 ijms-24-13572-f001:**
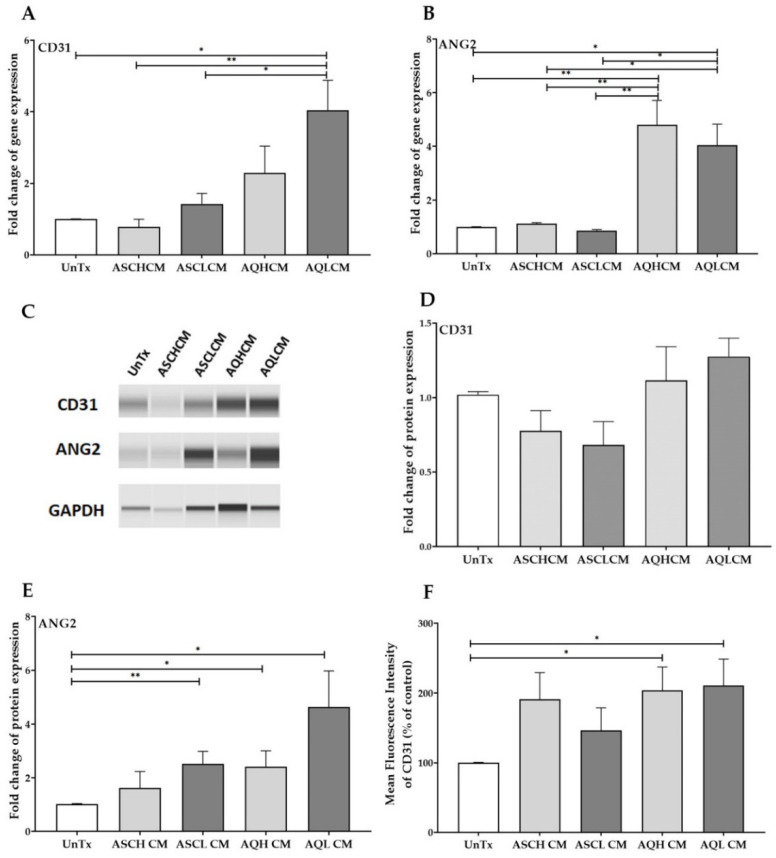
(**A**,**B**) qRT-PCR showed a significant increase in the expression of CD31 and ANG2 in HUVECs treated with pooled AQ CM (n = 3) in 2D monolayer culture. (**C**) Capillary Western blot (Jess) assay showing CD31, ANG2, and GADPH protein expression in an assembled gel-like image view. (**D**,**E**) Quantification of Western blot gels showing an increase in CD31 and ANG2 protein expression in HUVECs treated with pooled AQL CM (n = 3). (**F**) Quantification of mean fluorescence intensity (MFI) showing a significant increase in CD31 expression in HUVECs treated with AQH and AQL CM. The values are the mean SEM. * *p* < 0.05; ** *p* < 0.01. Abbreviations: UnTx: HUVECs untreated; ASCH CM and ASCL CM: HUVECs treated with CM collected from healthy and lipedema ASCs, respectively; AQH CM and AQL CM: HUVECs treated with CM collected from adipocytes differentiated from ASCH and ASCL, respectively.

**Figure 2 ijms-24-13572-f002:**
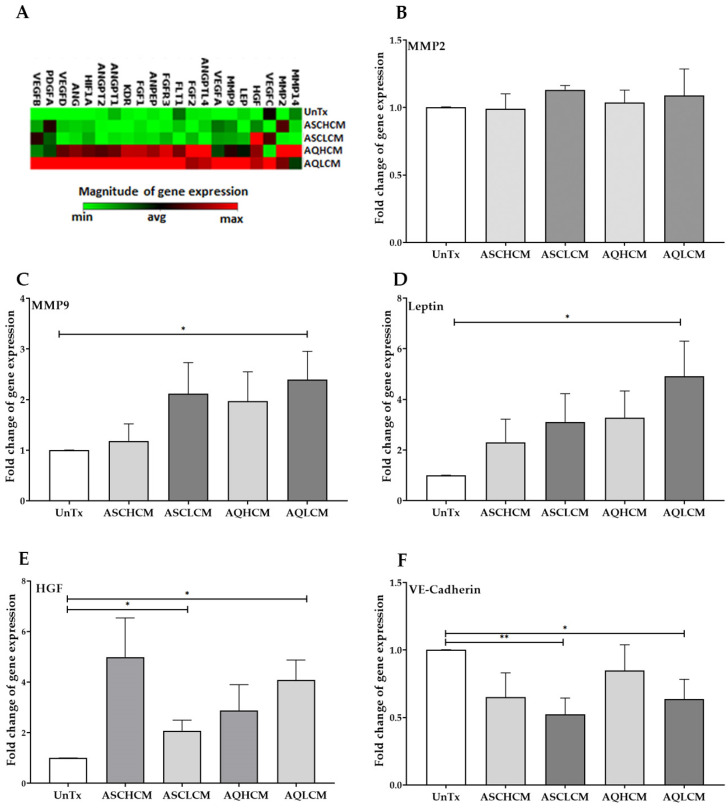
(**A**) Clustergram of angiogenetic RT2PCR array showing an increased expression (red squares) in the angiogenic markers in HUVECs treated with pooled AQH and AQL CM compared to control cells (green squares). (**B**–**E**) qRT-PCR showed no change in the expression of MMP2 (**B**), a significant increase in MMP9, leptin, and HGF expression in HUVECs treated with AQL CM (single cells, n = 5). (**F**) qRT-PCR showed a significant decrease in VE-cadherin expression in HUVECs treated with ASCH and AQL CM (single cells; n = 5). The values are the mean SEM. * *p* < 0.05; ** *p* < 0.01. Abbreviations: UnTx: HUVECs untreated; ASCH CM and ASCL CM: HUVECs treated with CM collected from healthy and lipedema ASCs, respectively; AQH CM and AQL CM: HUVECs treated with CM collected from adipocytes differentiated from ASCH and ASCL, respectively. MMP: matrix metallopeptidase; VEGF: vascular endothelial growth factor; HGF: hepatocyte growth factor, LEP: leptin; FGF: fibroblast growth factor; FGFR: fibroblast growth factor; ANG, ANGPT: angiopoietin; ANGPTL4: angiopoietin-like 4; FLT1/VEGFR1: FMS related receptor tyrosine kinase 1; PDGF-A: platelet-derived growth factors.

**Figure 3 ijms-24-13572-f003:**
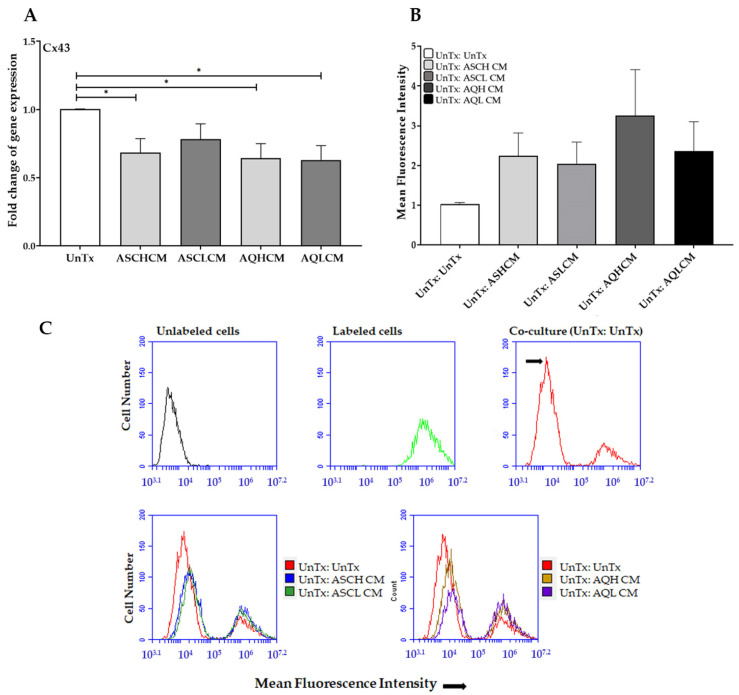
(**A**) qRT-PCR showed a significant decrease in the expression of Cx43 in treated HUVECs with CM (single cells, n = 4). (**B**) Histogram demonstrates the intercellular communication increase in treated HUVECs with CM as measured by a dye transfer assay (single cells, n = 3). (**C**) Homo-cellular gap junctional intercellular communication established between HUVECs, a representative flow cytometry graph where the shift in MFI is shown following co-culture of unlabeled HUVECs with calcein-labeled HUVECs (UnTx: UnTx). The arrow represents the shift in the MFI of unlabeled cells. The values are the mean SEM. * *p* < 0.05. Abbreviations: UnTx: HUVECs untreated; ASCH CM and ASCL CM: HUVECs treated with CM collected from healthy and lipedema ASCs, respectively; AQH CM and AQL CM: HUVECs treated with CM collected from adipocytes differentiated from ASCH and ASCL, respectively.

**Figure 4 ijms-24-13572-f004:**
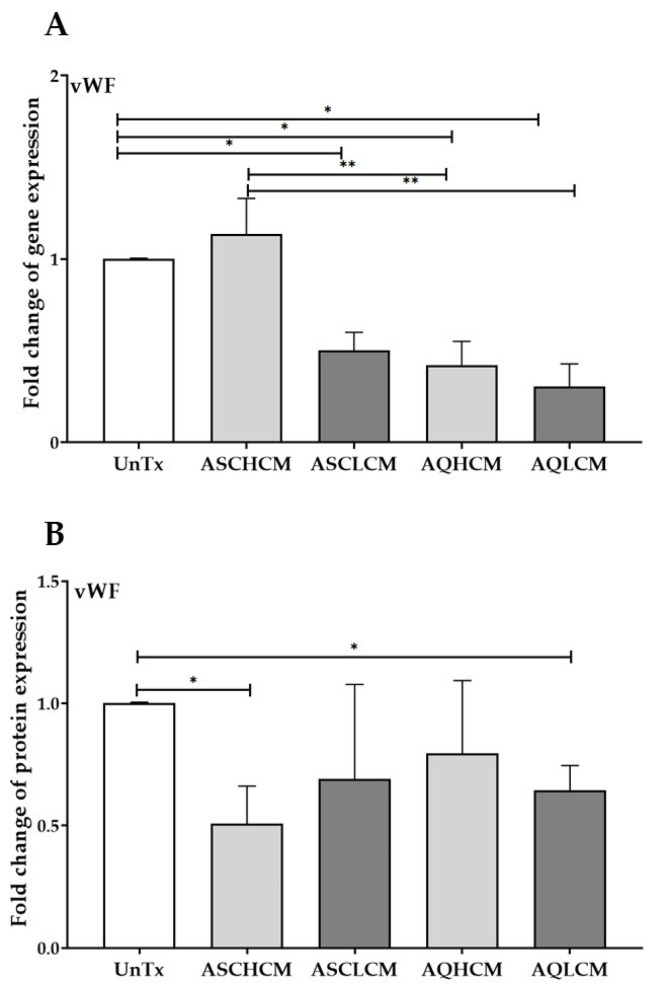
(**A**) qRT-PCR showed a significant decrease in the expression of vWF in treated HUVECs with CM (single cells, n = 5). (**B**) Quantifying Western blot gels shows decreased vWF expression in HUVECs treated with ASCH and AQL CM (single cells, n = 5). The values are the mean SEM. * *p* < 0.05; ** *p* < 0.01. Abbreviations: UnTx: HUVECs untreated; ASCH CM and ASCL CM: HUVECs treated with CM collected from healthy and lipedema ASCs, respectively; AQH CM and AQL CM: HUVECs treated with CM collected from adipocytes differentiated from ASCH and ASCL, respectively.

**Figure 5 ijms-24-13572-f005:**
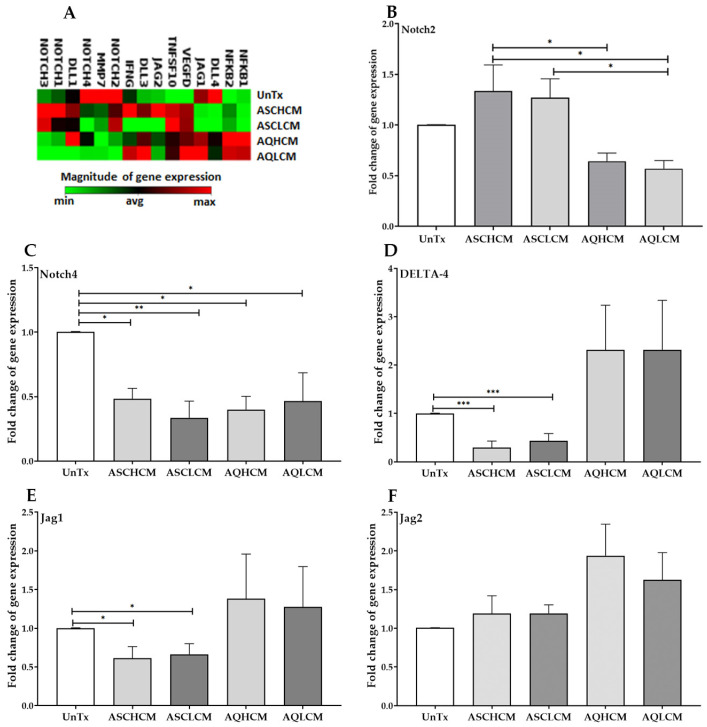
(**A**) Clustergram of NOTCH RT2PCR array showing an increased expression of inflammatory markers (red squares) and decrease in NOTCH genes and its ligands (green squares) in HUVECs treated with pooled AQH and AQL CM compared to control cells. (**B**–**F**) qRT-PCR showed a significant decrease in NOTCH2, NOTCH4, DELTA-4, and Jag1 (**B**–**E**) in HUVECs treated with CM (single cells, n = 5), but no significant change in the expression of Jag2 (**F**). The values are the mean SEM. * *p* < 0.05; ** *p* < 0.01; *** *p* < 0.001. Abbreviations: UnTx: HUVECs untreated; ASCH CM and ASCL CM: HUVECs treated with CM collected from healthy and lipedema ASCs, respectively; AQH CM and AQL CM: HUVECs treated with CM collected from adipocytes differentiated from ASCH and ASCL, respectively; NF-κB: nuclear factor kappa-light-chain-enhancer of activated B cells; DLL4: DELTA; IFNG: interferon gamma; MMP7: matrix metallopeptidase 7; TNFSF10: TNF Superfamily Member 10.

**Figure 6 ijms-24-13572-f006:**
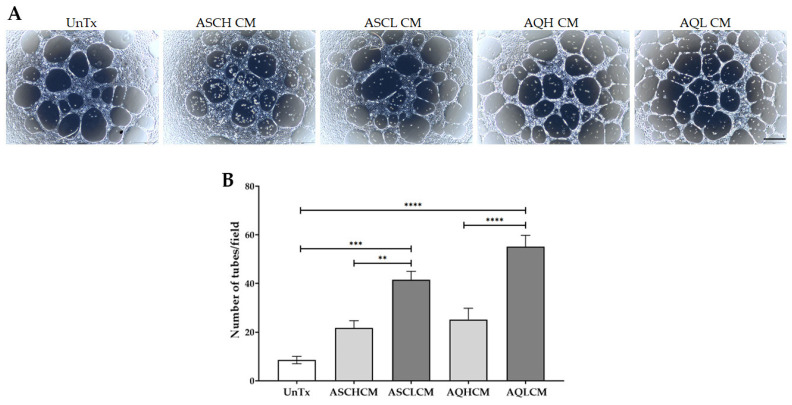
Matrigel tube formation assay. (**A**) Representative images of HUVEC tube formation (scale bar, 500 µm). (**B**) Quantification of tube length showing a significant increase in HUVECs treated with ASL, AQH, and AQL CM (single cells n = 3). The values are the mean SEM. ** *p* < 0.01; *** *p* < 0.001; **** *p* < 0.0001. Abbreviations: UnTx: HUVECs untreated; ASCH CM and ASCL CM: HUVECs treated with CM collected from healthy and lipedema ASCs, respectively; AQH CM and AQL CM: HUVECs treated with CM collected from adipocytes differentiated from ASCH and ASCL, respectively.

**Figure 7 ijms-24-13572-f007:**
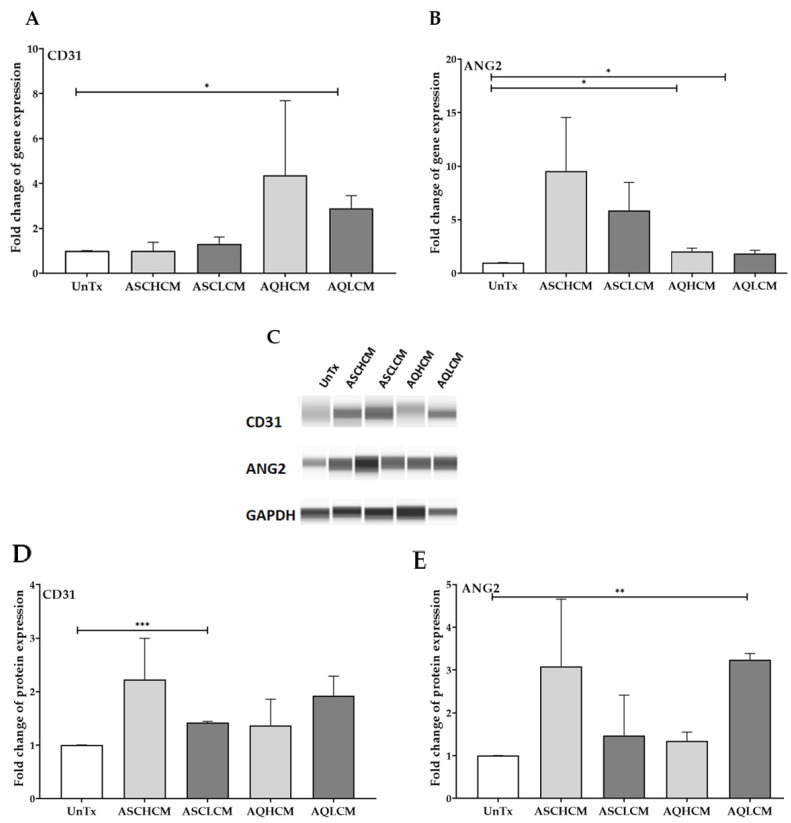
(**A**,**B**) qRT-PCR showing a significant increase in expression of CD31 and ANG2 in HUVECs treated with pooled AQ CM (n = 3) in 3D culture. (**C**) Western blot analysis showing CD31, angiopoietin 2, and GADPH protein expression in an assembled gel-like image view. (**D**,**E**) Quantification of Western blot gels showing an increase in CD31 and angiopoietin protein expression in HUVECs treated with ASC and AQ CM (n = 3). The values are the mean SEM. * *p* < 0.05; ** *p* < 0.01; *** *p* < 0.001. Abbreviations: UnTx: HUVECs untreated; ASCH CM and ASCL CM: HUVECs treated with CM collected from healthy and lipedema ASCs, respectively; AQH CM and AQL CM: HUVECs treated with CM collected from adipocytes differentiated from ASCH and ASCL, respectively.

**Table 1 ijms-24-13572-t001:** Characteristics of healthy and lipedema patients.

Characteristics	Healthy	Lipedema
**For pooled cells**		
**N**	5	5
**Sex**	Female	Female
**Age**	45 ± 6.42	49.8 ± 3.97
**BMI**	29.2 ± 2.77	29.9 ± 3.34
**Stage 2**	-	100%
**For single cells**		
**N**	5	5
**Sex**	Female	Female
**Age**	45 ± 6.89	42 ± 10
**BMI**	27.2 ± 2.51	30.7 ± 3.2
**Stage 2**	-	100%

**Table 2 ijms-24-13572-t002:** List of primers used for qRT-PCR.

Name	Forward (5′-3′)	Reverse (5′-3′)
Cx43	CTTCACTACTTTTAAGCAAAAGAG	TCCCTCCAGCAGTTGAG
ZO-1	CAGCCGGTCACGATCTCCT	GTGATGGACGACACCAGCG
HGF	ATTGCCCTATTTCTCGTTGTG	GCATTTCTCATCTCCTCTTCC
JAG-1	GACTCATCAGCCGTGTCTCA	TGGGGAACACTCACACTCAA
JAG-2	TCTGCCTTGCTACAATGGTG	GCGATACCCGTTGATCTCAT
vWF	TCTTCCAGGACTGCAACAAG	TCCGAGATGTCCTCCACATA
VEGF	AGGCCCACAGGGATTTTCTT	ATCAAACCTCACCAAGGCCA
MMP-2	TTGACGGTAAGGACGGACTC	ACTTGCAGTACTCCCCATCG
MMP-9	TTGACAGCGACAAGAAGTGG	GCCATTCACGTCGTCCTTAT
Leptin	GAAGACCACATCCACACACG	AGCTCAGCCAGACCCATCTA
ANGPT2	TGCCACGGTGAATAATTCAG	TTCTTCTTTAGCAACAGTGGG
GAPDH	CGCTGAGTACGTCGTGGAGTC	GCAGGAGGCATTGCAGATGA
NOTCH1	CACTGTGGGCGGGTCC	GTTGTATTGGTTCGGCACCAT
NOTCH2	AATCCCTGACTCCAGAACG	TGGTAGACCAAGTCTGTGATGAT
NOTCH3	TGACCGTACTGGCGAGACT	CCGCTTGGCTGCATCAG
NOTCH4	TAGGGCTCCCCAGCTCTC	GGCAGGTGCCCCCATT
DELTA-4	CGTCTGCCTTAAGCACTTCC	GAAATTGAAGGGCAGTTGGA
VE-cadherin	GACTTGGCATCCCATTGTCT	ACCCCCACAGGAAAAGAATC

## Data Availability

Not applicable

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
