# Peer review of "Enhanced Angiogenesis in HUVECs Preconditioned with Media from Adipocytes Differentiated from Lipedema Adipose Stem Cells In Vitro"

_ijms, 2023, doi:10.3390/ijms241713572_

Round 1

Reviewer 1 Report

Comments to authors-2023-08-08

Manuscript ID: ijms-2564723

Thank you very much to give me an opportunity to review this manuscript.

The paper described endothelial cell bioassay using HUVEC with conditioned media purified from adipocytes differentiated from lipedema adipose derived stem cells.

The experimental model is an interesting to explore factors from production from adipocytes purified from patients with lipedema to affect bioactivity of vascular endothelial cells.

#1. Results representing in figure 1B and figure 7B seemed to be contradicting. How do authors explain about these results? Was the difference of results due to culture condition between 2D and 3D? Usually tube formation was frequently observed in 3D culture system rather than 2D. I cannot understand when I read through the whole manuscript.

#2. I cannot understand how to purify AQH and AQL. How did authors discriminate between these cells? Was AQH derived from normal volunteer?

#3. How did authors explain significance of downregulation of NOTCH signal pathway in HUVEC used in the current study?

Author Response

We thank the reviewer for their astute comments and suggestions, which we have addressed in the revised manuscript. All concerns have been discussed. In the following sections, we have addressed the comments specific to each reviewer.

Response to Reviewer 1

Thank you very much to give me an opportunity to review this manuscript.

The paper described endothelial cell bioassay using HUVEC with conditioned media purified from adipocytes differentiated from lipedema adipose derived stem cells.

The experimental model is an interesting to explore factors from production from adipocytes purified from patients with lipedema to affect bioactivity of vascular endothelial cells.

#1. Results representing in figure 1B and figure 7B seemed to be contradicting. How do authors explain about these results? Was the difference of results due to culture condition between 2D and 3D? Usually, tube formation was frequently observed in 3D culture system rather than 2D. I cannot understand when I read through the whole manuscript.

Response: Figure 1 and Figure 7 show the expression of CD31 and ANG2 in 2D monolayer culture and 3D culture, respectively. The main difference between Figure 1 and 7 is the increase in expression of ANG2 in HUVECS treated with ASC CM in 3D culture compared to 2D cultures, which is due to the difference in culture conditions; However, both figures showed a significant increase in CD31 and ANG2 in HUVECS treated with AQ CM compared to untreated HUVECs which is the focus of this manuscript.

HUVECs tube formation assay is a widely accepted assay of in vitro angiogenesis. Studies have shown that HUVECs form endothelial tubes in 2D and 3D culture models [1-4]. In this manuscript, we aimed to show tube formation in 2D cultures. We apologize for the confusion and added “2D or 3D cultures” to the paragraphs. We hope that this clarifies the issue.

#2. I cannot understand how to purify AQH and AQL. How did authors discriminate between these cells? Was AQH derived from normal volunteer?

Response: AQH refers to adipocytes differentiated from healthy “normal”, non-lipedema, and adipose-derived stem cells (ASC H). These cells were characterized in our previous publications [5, 6]. Briefly, ASCs from healthy (ASC H) and lipedema (ASC L) cell lines were grown in DMEM/F12 media. Media was changed every three days until confluence. After this, the media was replaced with an adipogenic differentiation- medium to induce differentiation of  the cells, which we referred to as AQ plates. On Day 21, a total of 50mL of conditioned media (CM) was collected from healthy (AQH CM) and lipedema (AQL CM) plates and used in this study.

#3. How did authors explain significance of downregulation of NOTCH signal pathway in HUVEC used in the current study?

Response: NOTCH signaling plays a significant role in regulating the angiogenic pathway. Studies have shown that the downregulation of NOTCH expression leads to the induction of angiogenesis [7, 8]. Our data show a decrease in the expression of NOTCH 1-4 and its receptors Delta 1 and 4 and an increase in CD31 and ANG2 expression as well as an increase in tube formation HUVECs treated cells; thus, taken together, these data confirm that downregulation of Notch induces angiogenesis in HUVECs. Therefore, targeting NOTCH signaling will help researchers develop potential treatments to control/inhibit angiogenesis in Lipedema.

References:

  1. DeCicco-Skinner, K.L., et al., Endothelial cell tube formation assay for the in vitro study of angiogenesis. J Vis Exp, 2014(91): p. e51312.
  2. Andrée, B., et al., Formation of three-dimensional tubular endothelial cell networks under defined serum-free cell culture conditions in human collagen hydrogels. Scientific Reports, 2019. 9(1): p. 5437.
  3. Perugini, V. and M. Santin, A Substrate-Mimicking Basement Membrane Drives the Organization of Human Mesenchymal Stromal Cells and Endothelial Cells Into Perivascular Niche-Like Structures. Front Cell Dev Biol, 2021. 9: p. 701842.
  4. Majewska, A., et al., Endothelial Cells as Tools to Model Tissue Microenvironment in Hypoxia-Dependent Pathologies. International Journal of Molecular Sciences, 2021. 22(2): p. 520.
  5. Al-Ghadban, S. et al., Increase in Leptin and PPAR-γ Gene Expression in Lipedema Adipocytes Differentiated in vitro from Adipose-Derived Stem Cells. Cells, 2020. 9(2).
  6. Al-Ghadban, S., et al., 3D Spheroids Derived from Human Lipedema ASCs Demonstrated Similar Adipogenic Differentiation Potential and ECM Remodeling to Non-Lipedema ASCs In Vitro. Int J Mol Sci, 2020. 21(21).
  7. Benedito, R., et al., Loss of Notch signalling induced by Dll4 causes arterial calibre reduction by increasing endothelial cell response to angiogenic stimuli. BMC Dev Biol, 2008. 8: p. 117.
  8. Oon, C.E. and A.L. Harris, New pathways and mechanisms regulating and responding to Delta-like ligand 4-Notch signalling in tumour angiogenesis. Biochem Soc Trans, 2011. 39(6): p. 1612-8.

Reviewer 2 Report

The original article authored by Dr. Al-Ghadban and colleagues, titled "Enhanced Angiogenesis in HUVECs Preconditioned with Media from Adipocytes Differentiated from Lipedema Adipose Stem Cells In Vitro," explores the impact of conditioned media (CM) derived from adipose stem cells (ASCs) and adipose tissue from both healthy individuals and patients with lipedema on protein and gene expression in endothelial cells. 

While the paper is well-written, there are certain critical aspects that require further attention (Major revision):

-       Given that Lipedema primarily affects females, starting in puberty, it is plausible to consider it a hormone-sensitive pathology. Despite this, the authors have not addressed the potential influence of hormones in their analysis. Since endothelial cells possess estrogen receptors and can respond to hormones, it would be insightful for the authors to investigate estrogen receptor (ER) activation or downstream signaling in their study.

-       In the Methods and Materials section (M&M – lines 98-99), the authors mention that ASCs are derived directly from lipoaspirates of subcutaneous adipose tissue. However, the authors have not explained why they did not isolate endothelial cells and utilize them for RT2 profiler PCR assays or flow cytometry analysis. While this aspect is briefly touched upon in the Discussion section (lines 562-563), further elaboration and exploration of this approach would be beneficial.

Minor revision:

-       Are HUVECs used as pool or single batch sample? Please insert in M&M or in the figure legends.

-       Figure 1, 2, 3, 5, contain graphs without statistics. Please insert the correction.

-       Please provide (in supplementary data) the gating strategy for flow cytometry expreiments.

Author Response

Response to reviewers’ comments

We thank the reviewer for their astute comments and suggestions, which we have addressed in the revised manuscript. All concerns have been discussed. In the following sections, we have addressed the comments specific to each reviewer.

Response to Reviewer 2

The original article authored by Dr. Al-Ghadban and colleagues, titled “Enhanced Angiogenesis in HUVECs Preconditioned with Media from Adipocytes Differentiated from Lipedema Adipose Stem Cells In Vitro,” explores the impact of conditioned media (CM) derived from adipose stem cells (ASCs) and adipose tissue from both healthy individuals and patients with lipedema on protein and gene expression in endothelial cells. 

While the paper is well-written, there are certain critical aspects that require further attention (Major revision):

-       Given that Lipedema primarily affects females, starting in puberty, it is plausible to consider it a hormone-sensitive pathology. Despite this, the authors have not addressed the potential influence of hormones in their analysis. Since endothelial cells possess estrogen receptors and can respond to hormones, it would be insightful for the authors to investigate estrogen receptor (ER) activation or downstream signaling in their study.

Response: We agree with the reviewer’s suggestion that hormones play an essential role in the development and progression of lipedema. Studies have shown that estrogen also induces angiogenesis in HUVECs [1, 2]. Our next step will be to characterize endothelial cells (EC) isolated from healthy and Lipedema patients at the transcriptional and translational levels for the expression of ERs. In addition, we want to co-culture ECs and ASCs in the presence of 17β-Estradiol (E2) and determine the effect of both cellular and paracrine interactions on the angiogenic potential of ECs in vitro. This work will be part of a more extensive study investigating the role of estrogen in the pathophysiology of  Lipedema and identifying potential biomarkers that will prevent the progression of this disease.

-       In the Methods and Materials section (M&M – lines 98-99), the authors mention that ASCs are derived directly from lipoaspirates of subcutaneous adipose tissue. However, the authors have not explained why they did not isolate endothelial cells and utilize them for RT2 profiler PCR assays or flow cytometry analysis. While this aspect is briefly touched upon in the Discussion section (lines 562-563), further elaboration and exploration of this approach would be beneficial.

Response: Endothelial cells (CD31+ cells) represents ~10-20% of the stromal vascular fraction isolated from the lipoaspirate [3]. However, due to the limited availability of lipoaspirate samples from patients, the isolation and purification procedure of ECs was not feasible. Thus, we used HUVECs to get preliminary data.

Minor revision:

-       Are HUVECs used as pool or single batch sample? Please insert in M&M or in the figure legends.

Response: HUVECs used in this study are from pooled donors. It has been added to the Materials and Methods section.

-       Figure 1, 2, 3, and 5 contain graphs without statistics. Please insert the correction.

Response: The data shown in Fig 1D, Fig 2B, Fig 3B, Fig 5F is not statistically significant (p>0.05).

-       Please provide (in supplementary data) the gating strategy for flow cytometry experiments.

Response: We analyzed the data using BD Accuri C6 plus software for flow analysis. The dye transfer assay was calculated by dividing the mean fluorescent intensity (MFI) of M1 over the percentage of adhered fluorescent cell population after each co-culture. The figure has been added to the supplementary data.

References:

  1. Sobrino, A., et al., Estradiol stimulates vasodilatory and metabolic pathways in cultured human endothelial cells. PLoS One, 2009. 4(12): p. e8242.
  2. Morales, D.E., et al., Estrogen promotes angiogenic activity in human umbilical vein endothelial cells in vitro and in a murine model. Circulation, 1995. 91(3): p. 755-63.
  3. Saito, N., et al., Purification and characterization of human adipose-resident microvascular endothelial progenitor cells. Scientific Reports, 2022. 12(1): p. 1775.

Round 2

Reviewer 1 Report

Thank you for authors to respond my commets.

I understand that authors give me comments sincerely.

However, I would like to ask authors to correct figure legends in figure 1 and 7.

Still I could not understand the difference of figure 1B and 7B, although authors explain that those are different between 2D and 3D culture.

I feel confusing of figure 1B and 7B.

Otherwise, I am not willing to accept this manuscript.

Author Response

We have added 2D monolayer and 3D cultures to Figure 1 and Figure 7, respectively. We have also modified paragraph “2.6” explaining that the differences in ANG2 gene expression between 2D and 3D monolayer cultures is due to the different culture conditions.

Here are our explanations:

Studies have shown that cells cultured in 2D monolayer have differences in gene and protein expression compared to cells grown in 3D cultures, which is due to the following differences:

  1. Cellular matrix: In 2D culture, cells adhere to the tissue culture plastic and grow flat and proliferate, whereas in 3D cultures, cells aggregate together due to the non-adherent to the plastic, and the primary cells that proliferate are the outer layer.
  2. Microenvironment: In 2D culture, the cellular microenvironment is static with limited interactions between cells, whereas, in 3D cultures, the microenvironment is dynamic, allowing more physiologically relevant interactions between cells and cell produced-ECM, thus providing an ideal condition to study physiological functions more similar to that of an in vivo environment.
  3. Effect of treatment: In 2D culture, cells are exposed to a homogenous treatment (in our case, it is the conditioned media) in the tissue culture dish, whereas in 3D cell cultures, cells are exposed to varied amounts of treatment in the culture vessels.

Taking all these together, we hypothesize that the difference demonstrated in ANG2 gene expression between our 2D monolayer (Fig 1B) and 3D (Fig 7B) cultures is due to the abovementioned differences.

We hope that this clarifies the issue.

Thank you! 

Reviewer 2 Report

N/A

Author Response

Thank you! 

Round 3

Reviewer 1 Report

The current revised manuscript was corrected as I suggested.

I am finally satisfied with authors' response.